# Post-Vaccination and Post-Infection Immunity to the Hepatitis B Virus and Circulation of Immune-Escape Variants in the Russian Federation 20 Years after the Start of Mass Vaccination

**DOI:** 10.3390/vaccines11020430

**Published:** 2023-02-13

**Authors:** Fedor A. Asadi Mobarkhan, Victor A. Manuylov, Anastasia A. Karlsen, Vera S. Kichatova, Ilya A. Potemkin, Maria A. Lopatukhina, Olga V. Isaeva, Eugeniy V. Mullin, Elena P. Mazunina, Evgeniia N. Bykonia, Denis A. Kleymenov, Liubov I. Popova, Vladimir A. Gushchin, Artem P. Tkachuk, Anna A. Saryglar, Irina E. Kravchenko, Snezhana S. Sleptsova, Victor V. Romanenko, Anna V. Kuznetsova, Sergey A. Solonin, Tatyana A. Semenenko, Mikhail I. Mikhailov, Karen K. Kyuregyan

**Affiliations:** 1Department of Socially Significant Viral Infections, Russian Medical Academy of Continuous Professional Education, 125993 Moscow, Russia; 2Laboratory of Viral Hepatitis, Mechnikov Research Institute of Vaccines and Sera, 105064 Moscow, Russia; 3Gamaleya National Research Center for Epidemiology and Microbiology, 123098 Moscow, Russia; 4Scientific and Educational Resource Center for High-Performance Methods of Genomic Analysis, Peoples’ Friendship University of Russia (RUDN University), 117198 Moscow, Russia; 5Hospital of Infectious Diseases, 667003 Kyzyl, Russia; 6Department of Infectious Diseases, Kazan State Medical University of the Ministry of Healthcare of Russia, 420012 Kazan, Russia; 7Medical Institute, M.K. Ammosov North-Eastern Federal University, 677010 Yakutsk, Russia; 8Medical Faculty, Ural State Medical University, 620014 Yekaterinburg, Russia; 9Center for the Prevention and Control of AIDS and Infectious Diseases under Health Ministry of Khabarovsk Region, 680031 Khabarovsk, Russia; 10N.V. Sklifosovsky Research Institute for Emergency Medicine of the Moscow Health Department, 129090 Moscow, Russia; 11Medical Faculty, Belgorod State National Research University, 308015 Belgorod, Russia

**Keywords:** Hepatitis B virus (HBV), HBV seroprevalence, Hepatitis B epidemiology, Hepatitis B vaccination, HBV humoral immunity, HBV immune escape variants

## Abstract

A neonatal vaccination against the Hepatitis B virus (HBV) infection was initiated in Russia 20 years ago, with catch-up immunization for adolescents and adults under the age of 60 years launched in 2006. Here, we have assessed the humoral immunity to HBV in different regions of Russia, as well as the infection frequency following 20 years of a nationwide vaccination campaign. We have also evaluated the role of immune-escape variants in continuing HBV circulation. A total of 36,149 healthy volunteers from nine regions spanning the Russian Federation from west to east were tested for HBV surface antigen (HBsAg), antibodies to HBV capsid protein (anti-HBc), and antibodies to HBsAg (anti-HBs). HBV sequences from 481 chronic Hepatitis B patients collected from 2018–2022 were analyzed for HBsAg immune-escape variants, compared with 205 sequences obtained prior to 2010. Overall, the HBsAg detection rate was 0.8%, with this level significantly exceeded only in one study region, the Republic of Dagestan (2.4%, *p* < 0.0001). Among the generation vaccinated at birth, the average HBsAg detection rate was below 0.3%, ranging from 0% to 0.7% depending on the region. The anti-HBc detection rate in subjects under 20 years was 7.4%, indicating ongoing HBV circulation. The overall proportion of participants under 20 years with vaccine-induced HBV immunity (anti-HBs positive, anti-HBc negative) was 41.7% but below 10% in the Tuva Republic and below 25% in the Sverdlovsk and Kaliningrad regions. The overall prevalence of immune-escape HBsAg variants was 25.2% in sequences obtained from 2018–2022, similar to the prevalence of 25.8% in sequences collected prior to 2010 (*p* > 0.05). The population dynamics of immune-escape variants predicted by Bayesian analysis have remained stable over the last 20 years, indicating the absence of vaccine-driven positive selection. In contrast, the wild-type HBV population size experienced a rapid decrease starting in the mid-1990s, following the introduction of mass immunization, but it subsequently began to recover, reaching pre-vaccination levels by 2020. Taken together, these data indicate that it is gaps in vaccination, and not virus evolution, that may be responsible for the continued virus circulation despite 20 years of mass vaccination.

## 1. Introduction

The elimination of Hepatitis B as a public health threat by 2030 is one of the goals of the World Health Organization (WHO) within the global health sector strategy on viral hepatitis. Along with a reduction in mortality associated with the Hepatitis B virus (HBV), the impact target to be achieved as a result of this strategy is a reduction in infection incidence to the rates equivalent to 1% prevalence of the virus surface antigen (HBsAg) among children in 2020, and to 0.1% by 2030. Vaccination, primarily universal newborn vaccination with the first vaccine dose given at birth, is the main tool for preventing HBV infection, and one of the key services of the elimination strategy, along with blood safety, safe injections, and coverage through diagnostics and the treatment of chronic HBV infection [1]. According to the WHO strategy, childhood vaccine coverage, including all three vaccine doses, should be at least 90% in 2020, and coverage through services targeting the prevention of mother-to-child transmission of HBV, including vaccination at birth, the screening of all pregnant women, and the antiviral treatment for HBV-infected pregnant women [2] should be at least 90% by 2030 [1].

A universal neonatal HBV vaccination program involving three vaccine doses, with the first dose given at birth, has been implemented in Russia since 1998. In addition, an HBV vaccination campaign targeting adolescents and adults under the age of 60 years was launched in 2006. By 2013, the reported HBV vaccination coverage was 97% in children under 18 and 72% in adults [3]. As a result of these efforts, the reported acute Hepatitis B incidence rates dropped to 0.35 cases per 100,000 in 2020, and to 0.28 cases per 100,000 in 2021, compared to 35–43 per 100,000 observed in the pre-vaccination period [4]. However, the data on HBV prevalence in the general population, as well as on herd immunity to the virus in Russia, are fragmentary. Previously, we have demonstrated ongoing HBV circulation in two regions of Russia despite 20 years of a universal vaccination program. This was presumably associated with gaps in vaccination and/or suboptimal vaccination timeliness [5]. Theoretically, another factor contributing to enduring HBV circulation is the existence of immune-escape variants of the virus that are able to evade the vaccine-associated immune response, and thus may have a selective advantage in the context of universal vaccination [6]. A number of amino acid (aa) substitutions within HBsAg, mainly within its major immunogenic domain known as “a” determinant (aa positions 124 to 147), were reported to be associated with breakthrough infections after HBV vaccination or human HBV immunoglobulin prophylaxis [7]. Moreover, studies from Taiwan have shown that the prevalence of immune-escape variants of HBV increased during the early phase of the nationwide vaccination program and remained stable thereafter [8]. Although immune-escape variants do not appear to pose a serious threat to the effectiveness of vaccination programs, the monitoring of such virus variants during long-term immunization campaigns is an important task, given the limited data available on dynamic changes in their prevalence in most parts of the world. The aim of this study was to assess vaccine- and infection-induced immunity to HBV, as well as infection prevalence after 20 years of a nationwide vaccination campaign, and to evaluate the role of immune-escape variants in continuing HBV circulation. For this purpose, we conducted a serosurvey in different regions of Russia and analyzed the prevalence and spectrum of HBV immune-escape mutations, using as a reference point the data obtained in similar study by Klushkina et al. [9] conducted in 2008, 10 years after the implementation of universal vaccination for newborns. The motivation for this study was the lack of data on the changes in frequency of HBV infection following 20 years of mass neonatal vaccination in most regions of the Russian Federation that have different levels of registered incidence, as well as the need to understand the reasons underlying the persistence of HBV circulation. To address this question, we applied the novel approach combining the HBV serosurvey with an analysis of HBV molecular evolution under the pressure of mass vaccination. We assessed the age-specific rates of HBV surface antigen (HBsAg), antibodies to HBV core antigen (anti-HBc), and antibodies to HBsAg (anti-HBs) in healthy volunteers from different parts of the country, together with changes in the prevalence rates of HBV immune-escape variants using viral genomic sequences collected in the same territories in different years. Next, we applied Bayesian analysis to estimate the main parameters of HBV population dynamics, such as the effective number of infections and the reproduction number separately for the wild-type strains and immune-escape variants identified in this study.

## 2. Materials and Methods

### 2.1. Serum Samples

All of the serum samples taken from healthy volunteers and Hepatitis B patients tested in this study are shown in Figure 1, with their geographic origin also indicated. Since the annual rates of reported incidence of chronic Hepatitis B vary greatly among the regions of Russia, the 11-year average incidence rates in the regions studied from 2010–2021 are indicated with a colored bar in Figure 1.

In total, 36,149 healthy volunteers from nine regions spanning the Russian Federation from west to east were recruited for the HBV serosurvey from 2019–2020. The population sample size was calculated with the chosen power (80%) and confidence level (95%) [10] for the known size of the population in the study regions and the data on HBsAg and anti-HBc antibody prevalence in Russia obtained in previous years [9]. The subjects of the study were males and females between 0 and 95 years of age, all matching the following inclusion criteria: apparently healthy with no symptoms of acute disease at the time of enrollment in the study (either self-reported or parent-reported), and permanently residing in the study regions. Exclusion criteria were liver disease, acute illnesses, a body temperature over 37.1 °C, or any surgery, blood transfusion, or treatment with blood products within the three months prior to enrollment in the study. The study was made up of eight age groups, from children aged 1–14 years to senior citizens aged over 60 years (0–9, 10–14, 15–19, 20–29, 30–39, 40–49, 50–59, and ≥ 60 years). The mean population sample size in each age group was 295 individuals (51–3106). The mean male/female ratio was 1:1.3 and varied between 1:0.8 and 1:1.5 depending on the age group.

To study HBV genetic diversity and the prevalence of immune-escape variants, serum samples from 633 chronic Hepatitis B patients living in four regions (Moscow Region, Tuva Republic, Khabarovsk Region, and Yakutia) were collected (Figure 1). Samples from Yakutia were included in this study because the HBV serosurvey among healthy volunteers was performed recently in this region using the same study design that demonstrated ongoing HBV circulation in the vaccinated generation [5].

The entire study was conducted in accordance with the principles set out in the World Medical Association Declaration of Helsinki regarding ethical medical research involving human subjects. Informed written consent was obtained from all participants or their parents (for subjects under 15 years). The study design was approved by the Independent Interdisciplinary Ethics Committee for the Ethical Review of Clinical Research, Moscow, Russia (Approval No. 17 dated 16 November 2019) and by the Ethics Committee of the Mechnikov Research Institute for Vaccines and Sera in Moscow, Russia (Approval #1 dated 28 February 2018).

All serum samples obtained from healthy volunteers and Hepatitis B patients were coded and aliquoted, and all aliquots were stored at −70 °C until testing.

### 2.2. HBV Testing

All serum samples from healthy volunteers were tested for HBV surface antigen (HBsAg), antibodies to HBV core antigen (anti-HBc), and antibodies to HBsAg (anti-HBs) using commercially available enzyme-linked immunosorbent assay (ELISA) kits (Vector-Best, Novosibirsk, Russia) in accordance with the manufacturer’s instructions. The sensitivity of the HBsAg ELISA kit used in this study (HBsAg-EIA-BEST kit) was 0.01 IU/mL. The diagnostic sensitivity of the anti-HBc test was 100% (95% CI: 98.5–100%) according to manufacturer’s data. The limit of detection of anti-HBs test was 2 mIU/mL with linear range 10 to 1000 mIU/mL according to kit specifications. Samples were considered anti-HBs positive if the antibody concentration was ≥10 mIU/mL.

Viral nucleic acids were extracted from the patients’ sera using the QIAamp Viral RNA Mini Kit (QIAGEN, Hilden, Germany). HBV DNA testing was performed in an in-house PCR assay with nested primers for overlapping regions of S and P genes adapted from A. Basuni and W. Carman [11] using the Tersus Plus PCR kit (Eurogen, Moscow, Russia) containing high-fidelity polymerase. The assay’s detection limit was approximately 50 IU/mL, based on the results of serial dilution testing for PCR standards with a known viral load. The resulting 713 bp amplicons were extracted from the gel using a QIAquick Gel Extraction Kit (QIAGEN, Hilden, Germany) and subjected to direct Sanger sequencing on the 3130 Genetic Analyzer (ABI, Foster City, CA, USA) automatic sequencer using the BigDye Terminator v3.1 Cycle Sequencing Kit in accordance with the manufacturer’s protocol. All0 HBV sequences obtained in this study were deposited in the GenBank database under accession numbers OP956209–OP956545, OP972245–OP972274, OM025252, OM304326, and OQ026177–OQ026196.

### 2.3. HBV Sequence Analysis

Alignment of the nucleotide and predicted amino acid HBV sequences was performed using MEGA 7.0.18 [12]. The following substitutions associated with immune escape according to published data [7,11] were searched for in predicted aa sequences of HBsAg: T116N, T118A, P120S/E/T, T123N, T/I126A/N/S, Q129H/R, G130R, T131I, M133L, K141E, P142S, D144A/E/G, and G145R/A/K. To obtain two time points for the analysis of immune-escape variants, sequences obtained in this study were supplemented with 205 HBV sequences isolated by the authors in the same regions prior to 2010. None of the sequences obtained prior to 2010 were isolated from the same patients that participated in the current study, i.e., datasets “before 2010” and “after 2010” did not contain sequences from the same individuals. This subset included 55 sequences from Moscow Region (GenBank accession numbers OP972386–OP972440), 76 from Tuva (GenBank accession numbers OP972442–OP972472, OL449629, OL449643, OL449646, OL449649, OL449651, OL449654, OL449656, OL449657, OL449661, OL449662, and OQ026197–OQ026231), 58 from Yakutia (GenBank accession numbers OL771256, OL771260, AY653781–AY653783, AY653787, AY653790, AY653791, AY653796, AY653799, EU594390–EU594395, EU594433, OM025238–OM025253, OQ026232–OQ026244, and OQ204113–OQ204124), and 16 from Khabarovsk Region (GenBank accession numbers OP972370–OP972385).

Furthermore, to increase the sample of modern HBV sequences, the dataset was supplemented with 95 HBV sequences isolated in Yakutia from 2018–2019 from patients with HBV monoinfection (GenBank accession numbers: OP972275–OP972369). The frequency of clinically significant polymorphisms in HBsAg was calculated using Microsoft Office Excel.

HBV serotype was predicted based on the aa residues located at positions 122, 127, 134, 159, and 160 of the small S protein, which determine one or another HBsAg determinant [13]. HBV genotyping was performed through the phylogenetic analysis of a 640-nucleotide fragment of the S-gene (nucleotide positions 157–796 by reference sequence NC_003977.2) with the references recommended by the International Committee on Taxonomy of Viruses (ICTV) [14], suggested for HBV sub-genotype designation [15] and sequences from GenBank with known year and country of isolation, including ancient ones, in a total of 944 sequences using the Maximum Likelihood (ML) method in the PhyML-3.1 software. In the resulting datasets, the ratio between Russian isolates and sequences from the databases was at least 1:1.48.

Skyline methods of Bayesian analysis were used to reconstruct the evolutionary dynamics of wild-type HBV and immune-escape variants. Prior to conducting Bayesian analysis, we checked for the presence of genetic changes between the sampling time points in the HBV dataset based on the tree constructed by the ML method using TempEst v.1.5 software and observed a linear regression curve (Appendix A), indicating the existence of a temporal signal in the dataset.

To obtain plots, trees were built separately using only sequences of wild-type HBV or immune-escape variants from the study regions. Two different methods, Bayesian SkyGrid reconstruction (the values of effective number of infections) and Birth–Death Skyline analysis (reproduction number, Re), were used. For all datasets, the main parameters for both models were taken from calculations based on the tree that was built previously [16]. The Bayesian SkyGrid reconstruction was used with the Tree Prior parameter defined as equal to 50 and the final time point of 100 years before the most recent sampling performed using BEAST v1.10.4 software.

A Birth–Death Skyline analysis was performed with previously selected parameters [16], using the BEAST2 software. In all cases, the length of the Markov Chain Monte Carlo (MCMC) was set to 100-million generations, ESS > 200. Tracer v1.7.2 was used for primary visualization of the results and construct plots, with the bdskytools R-package used for the final visualization.

### 2.4. Statistical Analysis

Data analysis was performed using graphpad.com. Statistical analysis includes assessing the significance of the differences in mean values between groups using Fisher’s exact test, or Chi-square with Yates correction for large values (significance threshold *p* < 0.05).

## 3. Results

### 3.1. Frequency of HBV Infection in General Population

The average detection rates of HBV markers among healthy volunteers in study regions are shown in Table 1.

The average HBsAg detection rate calculated based on these data was 0.8%, and the only study region where HBsAg detection rate significantly exceeded this level was the Republic of Dagestan (2.4%, two-tailed *p* < 0.0001, Chi-square with Yates’ correction). In three study regions (the Republic of Dagestan, the Kaliningrad Region, and the Tuva Republic), the average anti-HBc antibody detection rates significantly exceeded (two-tailed *p* < 0.05, Chi-square with Yates’ correction) the national average of 14.2%. 

The age-specific HBsAg and anti-HBc antibody detection rates are shown in Figure 2A,B, respectively, and in greater detail in Appendix A.

In children and adolescents under 20 years of age, i.e., in the generation born after the start of the mass vaccination program, regional HBsAg detection rates did not significantly exceed the national average, which was below 0.3% (Figure 2A). In four territories, Kaliningrad Region, Moscow Region, the Republic of Tatarstan, and Sverdlovsk Region, zero HBsAg detection rates were observed in the generation covered by vaccination at birth. In other study regions, HBsAg detection rates in subjects under 20 years of age were 0.1% in the Novosibirsk Region (4 out of 3270) and in the Khabarovsk Region (3 out of 2371), 0.2% in the St. Petersburg and Leningrad Region (1 out of 595) and in the Tuva Republic (1 out of 494), and 0.7% in the Republic of Dagestan (17 out of 2438). However, HBsAg detection rates were significantly higher in the 15–19 years age group in the Republic of Dagestan (1.3%, 7 out of 545) compared to all other regions (two-tailed *p* < 0.05, Chi-square with Yates’ correction). In adults, the national average of age-specific HBsAg detection rates were below 2%, but these were significantly exceeded in two regions. In the Republic of Dagestan, the HBsAg detection rate was significantly higher than the national average in all adult age groups, peaking at 6.9% in the 30–39 years age group. In the Tuva Republic, the peak HBsAg detection rate was observed in the 50–59 years age group (7.4%), although in other age groups in this region, HBsAg detection rates did not significantly exceed the national average.

National average age-specific anti-HBc antibody detection rates were 7.4% (845 out of 11,387) in participants under 20 years old, but they increased gradually with age, peaking at 33.7% in subjects over 60 years of age (Figure 2B). A similar pattern was observed in every study region, although in several regions, age-specific anti-HBc detection rates significantly exceeded the national average. In subjects under 20 years of age, the highest anti-HBc antibody detection rates were observed in the Kaliningrad Region (above 12%) and the Republic of Dagestan (above 11%), as well as in the Tuva Republic in children under 15 years old (above 15%) and in the Sverdlovsk Region in the 10–14 years age group (16.4%). Among adults, the highest anti-HBc antibody detection rates, exceeding 35%, were observed in the Republic of Dagestan and in the Tuva Republic in subjects over 30 years old (Figure 2B).

### 3.2. Vaccine-Induced Humoral HBV Immunity

The average proportion of participants are reactive for anti-HBs antibodies but non-reactive for anti-HBc antibodies, i.e., those who have only vaccine-induced HBV immunity, was 39.4% (Table 1). In four regions, the Republic of Dagestan, the Kaliningrad Region, the Tuva Republic, and the Sverdlovsk Region, this index was significantly below the national average (two-tailed *p* < 0.0001, Chi-square with Yates’ correction). The overall proportion of anti-HBs-positive, anti-HBc-negative samples in subjects under 20 years old was 41.7% (4746 out of 11,387). The age-specific rates of vaccine-induced HBV immunity in the study regions are shown in Figure 3A and in greater detail in Appendix A. These rates displayed a similar pattern in all the study regions: a gradual decline from the 0–9 years age group to the 15–19 years age group, with a subsequent burst in the 20–29 years age group, followed by a gradual decline (Figure 3A), which corresponds to the response to vaccination at birth and catch-up vaccination in adolescents and young adults born after 1998. However, in a few regions, the rates of vaccine-induced HBV immunity were significantly lower compared to the national average, both in children and in adults. In subjects under 20 years old, the lowest proportion of anti-HBs-positive, anti-HBc-negative individuals was observed in the Tuva Republic, with a figure below 10%. Vaccine-induced HBV immunity rates in subjects aged 10–19 years were also below 25% in the Sverdlovsk Region and in the Kaliningrad Region (Figure 3A). In young adults (20–29 years old), the lowest rates of vaccine-induced HBV immunity, although still reaching 50%, were also observed in the Kaliningrad Region, the Republic of Dagestan, the Sverdlovsk Region, and the Tuva Republic.

We also assessed the proportion of participants who were non-reactive for both anti-HBc and anti-HBs antibodies, i.e., who had no measurable vaccine- or infection-induced humoral HBV immunity. The overall proportion of such participants was 46.5% and was significantly higher in five out of the nine study regions (Kaliningrad Region, St. Petersburg and Leningrad Region, Moscow and Moscow Region, the Republic of Tatarstan, and Sverdlovsk Region), as shown in Table 1 (two-tailed *p* < 0.0001, Chi-square with Yates’ correction). Among subjects aged under 20 years, the rate of samples non-reactive for anti-HBc and anti-HBs was 48.5% (5518 out of 11,387). Age-specific proportions of participants without detectable humoral HBV immunity are shown for each study region in Figure 3B and in Appendix A. In general, these proportions were an inverse reflection of the level of vaccine-induced immunity, with the proportion of non-immune children and adolescents aged 10–19 years reaching an average of 40–50% and equaling or exceeding that among those over 40 years. The proportion of subjects without immunity to HBV was highest in the Tuva Republic, in the 10–14 years and 15–19 years age groups, amounting to 75.3% and 88.6%, respectively (Figure 3B). Furthermore, the proportion of subjects without immunity to HBV among those who should be vaccinated at birth was significantly below the national average in the Republic of Dagestan (0–9 years age group) and the Sverdlovsk Region (15–19 years age group) (Figure 3B).

### 3.3. Prevalence of HBV Genotypes, Serotypes, and Vaccine-Escape Variants

Subsequently, we assessed the genetic diversity of HBV against the background of 20 years of the mass vaccination program. Out of 633 serum samples from patients with chronic Hepatitis B, 386 samples yielded the amplified 713 nt DNA fragment, covering the region of interest within HBsAg and being sufficient for accurate HBV genotyping and serotyping. HBV genotype and serotype distribution, as well as the prevalence of immune-escape variants, are shown in Table 2 for two time points: sequences obtained during this study from 2018 to 2022 and sets of sequences collected prior to 2010. HBV genotype D (HBV-D), with its respective serotype ay, was predominant among samples studied at both time points, except for Yakutia, where Genotype A (HBV-A), with its respective serotype ad, was also prevalent. The proportion of HBV-A significantly decreased in sequences collected in Yakutia from 2018–2020 compared with sequences obtained in this region prior to 2010, from 56.9% to 40.7% (*p* = 0.0377, Fisher’s exact test), while proportions of HBV-D and HBV-C remained stable (*p* > 0.05, Fisher’s exact test). However, the proportions of ad and ay serotypes in Yakutia did not change significantly when compared between these two time points (*p* > 0.05, Fisher’s exact test) due to the preservation of the proportion of genotype HBV-C, which is also characterized by the ad serotype.

Phylogenetic analysis has shown that HBV sequences from Russia belonged to the sub-genotypes D1, D2, D3, A2, and C2 and did not form any regional clusters but were instead dispersed among sequences from other regions (Figure 4). In general, sequences carrying aa substitutions associated with immune escape (indicated in Figure 4 with red color and IE abbreviation) did not form any common clusters in the phylogenetic tree but were dispersed across the regional clusters together with wild-type sequences (indicated in Figure 4 as WT), highlighting the absence of a common ancestor for these mutant strains. The only exceptions were the sequences carrying T118A/V variants (indicated in Figure 4 with light blue color and IE abbreviation), which were almost all concentrated in a sub-genotype D2 cluster, and which represented the majority of the D2 sequences identified in this study. This clustering suggests that T118A/V may be a wild-type variant characteristic of sub-genotype D2, rather than a mutation selected by immune evasion.

All immune escape-associated substitutions in HBsAg identified in this study are shown in Table 2. In addition to aa substitutions with a known immune-escape phenotype, a number of substitutions to another amino acid residue was observed in the same clinically relevant positions (shown in Table 2 with plain text). Although the phenotype of such substitutions remains uncertain, we included these variants in the calculations of immune-escape prevalence rates due to their location in positions within HBsAg that are critical for evasion from vaccine-induced immunity. With the exception of the substitutions in Position 118 of HBsAg, which were abundant in HBV-D2 sequences from all the study regions, immune-escape variants were relatively rare, with detection rates varying in recent samples, although not to a statistically significant degree, from 6.1% in the Tuva Republic and Yakutia to 13.8% in the Khabarovsk Region (Table 2). In the sequences collected prior to 2010, no significant differences in the prevalence rates of immune-escape variants were observed between the study regions. When all the sequences carrying substitutions in amino acid positions associated with immune escape were analyzed, no significant changes in the prevalence of HBsAg variants were observed between the samples collected prior to 2010 and those collected from 2018–2022, with the exception of the Tuva Republic, where a significant increase was observed in recent years (*p* = 0.0353, Fisher’s exact test). However, when the highly prevalent T118A variant was excluded from the analysis, the prevalence of immune-escape variants remained stable over the years in this region as well (Table 2). When all the regional data were collated together, the overall prevalence of immune-escape HBsAg variants was 25.2% (121 out of 481) in the sequences obtained from 2018–2022, similar to the prevalence of 25.8% (53 out of 205) in sequences collected prior to 2010 (*p* = 0.8484, Fisher’s exact test). After the exclusion of the T118A/V variants from the analysis, these proportions remained similar, at 8.1% (39 out of 481) and 10.7% (22 out of 205), respectively (*p* = 0.3049, Fisher’s exact test).

Since comparable ratios between HBV-A and HBV-D genotypes and, respectively, between ad and ay serotypes were observed only in Yakutia, we were only able to analyze the differences in the prevalence of immune-escape variants between HBV genotypes for this study region. The prevalence was significantly higher in HBV-D sequences compared to HBV-A sequences (*p* = 0.0008, Fisher’s exact test) in the sequence set from 2018–2022, but not in the sequences collected prior to 2010. However, when the T118A/V variants were excluded from the analysis, no significant difference in the prevalence of immune-escape variants was observed between HBV genotypes.

Next, we separately analyzed the detection rates of HBV-D variants carrying aa substitutions in Position 118 that were prevalent in our study. When all the HBV-D sequences from the four study regions were analyzed together, these variants tended to be more prevalent in sequences obtained from 2018–2022 compared to sequences collected prior to 2010 (24.1% (83 out of 344) vs. 17.0% (28 out of 165), *p* = 0.085, Fisher’s exact test). However, when analyzed separately for each study region, a significant increase in the prevalence of T188 variants over time was observed only in the Tuva Republic (40.4% [19 out of 47] vs. 17.6% [13 out of 74], *p* = 0.0104, Fisher’s exact test).

It should be noted that all sera samples from patients with chronic Hepatitis B collected from 2018–2022 were reactive for HBsAg in the ELISA test used in this study, as HBsAg testing prior to HBV DNA amplification was part of the initial patient specimen screening in the lab. As such, none of the immune-escape-associated substitutions identified in this study resulted in HBsAg detection failure.

### 3.4. Comparison of Population Dynamics of Wild-Type HBV and Immune Escape Variants

To further study HBV circulation against the background of mass vaccination, we applied Bayesian analysis to estimate the effective number of infections and the reproduction number. The effective number of infections, or effective population size, is an abstract quantity that corresponds to the population size under an idealized model of reproduction. It represents the number of successful transmission events in a given time and provides a measure of genetic diversity and its fluctuations over time. The reproduction number (Re) is defined as the number of expected secondary infections of an infected individual. Re is closely related to the basic reproductive number (R0), but the latter additionally assumes a completely susceptible population, and thus the two quantities are equal only at the start of an epidemic outbreak. The Bayesian analysis was run separately for the wild-type strains and immune-escape variants identified in this study. The same task was performed separately for sequences carrying T118A/V substitutions. Thus, the same analysis was performed for three different sequence subsets that included wild-type strains, all immune-escape variants, or T118A/V variants only. Interestingly, both immune-escape variants and T118A/V variants have similar dynamics for the estimated effective number of infections (Figure 5A) and Re (Figure 5B), remaining stable over the past 20 years during the ongoing mass vaccination program. In contrast, the wild-type HBV population size showed a rapid decrease starting in the mid-1990s, following the introduction of mass immunization, but it subsequently began to recover, reaching pre-vaccination levels by 2020 (Figure 5A). The estimated Re values for wild-type HBV are significantly lower and more variable over time compared with those of the immune-escape and T118A/V variants. The Re dynamics of wild-type HBV showed two episodes of decrease, in the late 1990s and late 2000s. These dates correspond to the start of mass neonatal vaccination in 1998 and the start of catch-up vaccination in adults in 2006. However, the Re values of wild-type HBV recovered after each episode of decline (Figure 5B). Taken together, HBV immune-escape and T118A/V variants have similar population dynamics, which are not significantly affected by mass vaccination. Moreover, the population dynamics of wild-type HBV shows signs of restoration despite the ongoing vaccination program.

## 4. Discussion

This study had several goals. First, we assessed the changes in HBV circulation following 20 years of mass neonatal vaccination. To do this, we analyzed HBsAg detection rates across all age groups of the general population. The national average HBsAg detection rate dropped to 0.8%, with regional positivity rates ranging from 0.2% to 0.4% in regions in the European part of the country to 1.2% in the Tuva Republic and 2.4% in the Republic of Dagestan, where HBV remains endemic. Such a decline in average HBsAg detection rates seems to be substantial when compared with data from the 2008 serosurvey conducted 10 years after the start of mass vaccination, when the regional HBsAg prevalence rates varied from 1.6% in the Moscow Region and 1.2% in the Sverdlovsk Region to 3.2% in the Khabarovsk Region and 8.2% in the Tuva Republic [9]. The decline observed in HBsAg detection rates was most significant in the generation that had been vaccinated at birth as rates were below 1% in all regions; in six out of the nine regions studied, the detection rates in subjects under 20 years were no more than 0.1%. The decrease in HBsAg detection rates is most marked in the 15–19 years age group in the Tuva Republic, where it fell from 13.2% in 2008 [9] to zero prevalence in the current study. The observed transition from endemic to non-endemic HBV prevalence in response to national neonatal vaccination program is in line with data reported from different parts of the world, such as Gambia, Taiwan, Greece, and Alaska [17,18,19,20].

Furthermore, we assessed the anti-HBc antibody detection rates, which are indicative of the proportion of individuals who had been exposed to HBV and had presumably acquired infection-induced immunity. Despite the observed drop in HBsAg rates, the age-specific rates of reactivity to anti-HBc antibodies remained stable over the last decade in all age groups in the regions which were surveyed twice, currently and in 2008 [9], including the generation vaccinated at birth where positivity rates were approximately 10%. This latter figure is surprisingly high, contrasting, for instance, with the zero anti-HBc prevalence in the 12–14 years age group, and the 71% reduction among those aged 15–19 years reported in Australia a decade after the start of universal vaccination [21], or drop to below 1% observed in Italian recruits 30 years following the start of universal vaccination [22]. Notably, vaccination may not lead to a complete cessation of HBV circulation even in non-endemic countries, which is reflected by a non-zero anti-HBc antibody detection rates, but its prevalence demonstrates a significant reduction over time, as was reported in studies from the United States, the European Union, and China [23,24,25]. Our data on stable anti-HBc detection rates indicate ongoing HBV circulation in the vaccinated generation and the possibility of transient infection in vaccinated individuals that does not result in establishing HBsAg persistence. It should be noted that we did not test anti-HBc reactive samples for HBV DNA. As such, the proportion of latent, HBsAg-negative occult HBV infection (OBI) in anti-HBc-positive participants remains unknown. This constitutes a limitation of our study. Indeed, such an outcome of exposure to HBV in vaccinated individuals was evident both in the general population, contributing up to 13.8% of OBI prevalence in anti-HBc-positive adults vaccinated at birth [26], and in subsequently monitored groups of infants born to HBsAg-positive mothers [27]. Therefore, it is to be expected that a certain proportion of HBsAg-negative, anti-HBc-positive subjects from this study, the size of which has yet to be determined, has acquired OBI despite having been vaccinated at birth. This issue requires extensive study, as despite being almost fully protective against HBsAg-positive HBV infection, neonatal vaccination alone may be not sufficient to completely prevent HBV circulation and the formation of low-viremic OBI. The latter has clinical significance, as it may cause liver disease and lead to the development of liver cirrhosis and hepatocellular carcinoma (HCC) [28]. It also has implications for virus transmission, especially in transfusion scenarios [29,30].

Next, we assessed the role of immune-escape variants of HBV in maintaining virus circulation against the background of ongoing vaccination. We were not able to observe any increase in the prevalence of HBV variants carrying mutations in the HBsAg “a” determinant over time, at least over a 10-year interval. The detection rates observed in our study for such mutants (approximately 10%) were comparable to the 9–13% prevalence rates in HBV-A and HBV-D genotype sequences reported worldwide [31]. These data indicate the absence of positive selection of immune-escape variants of HBV during mass vaccination. The differences in the population dynamics of wild-type HBV and immune-escape variants observed in our study suggest the limited influence of vaccination on immune-escape mutants. However, no signs of an increase in the estimated population size of immune-escape variants were observed within the last 20 years, suggesting the absence of positive vaccine-driven selection. Furthermore, Bayesian analysis showed a recovery in the wild-type HBV population size following the initial decline after the start of mass vaccination, indicating a reduction in the impact of vaccination on HBV circulation. Such phylodynamics of wild-type HBV in Russia is significantly different from those described in other parts of the world. In general, there are two typical patterns of changes in HBV population size over time. The first pattern represents the exponential growth of viral population until mid-1990s with subsequent substantial decline, presumably in response to universal neonatal vaccination, as described for Genotype HBV-D in Iran and different viral genotypes in Japan [32,33]. The second pattern is characterized by the reaching plateau following the exponential growth in the middle of 20th century with subsequent stabilization or only slow decline since 1990s and was reported for genotypes HBV-A2 and HBV-C globally, or quasi-subgenotype HBV-B3 in Southeast Asia [34,35,36]. Our estimates of wild-type HBV population dynamics in Russia demonstrate that the rapid decrease in the viral population size in the mid-1990s, characteristic of the first pattern and indicative of success in control of HBV transmission, was subsequently replaced by the restoration of population size, reflecting a decrease in the effectiveness of anti-epidemic measures. Taken together, these data suggest that the limited effectiveness of HBV vaccination, indicated by the persistence of anti-HBc detection rates in subjects aged under 20 years, is associated with the ongoing circulation of wild-type virus strains, and not immune-escape variants.

The only excessively prevalent HBsAg polymorphism in the HBV-D genotype sequences in our study was T118A, located outside the “a” determinant, but within the HBsAg major hydrophilic region (MHR), the region between aa 100 and 160. Moreover, T118A/V was the only variant that tended to increase in prevalence with time. This variant was reported to be associated with the failure of Hepatitis B immunoglobulin (HBIG) prophylaxis [37] and, recently, with impaired binding to anti-HBs monoclonal antibodies. However, the latter was observed for the HBsAg variant carrying a double substitution, T118A/P127T [38]. As such, it is still unknown whether the T118A substitution is a true immune-escape variant, or just a natural polymorphism that has no impact on evasion from vaccine-induced immunity. Its predominance in sub-genotype D2 sequences observed in our study would support the second option. However, an analysis of the population dynamics of the T118A/V variant reconstructed using Bayesian analysis supports the immune-escape nature of this mutation. Indeed, the estimated population size and reproduction number values are very similar to those estimated for other immune-escape variants of HBV and are significantly different from those of the wild-type virus, showing the same potential advantage for these mutations during mass vaccination.

HBsAg mutants can also be an issue for HBV diagnostics, as they can influence the performance of commercial HBsAg assays [39]. However, all serum samples containing immune-escape variants were reactive for HBsAg in the ELISA test in this study, indicating the absence of evasion from diagnostics.

Taken together, our data suggest the insignificant role of HBsAg immune-escape variants in maintaining HBV circulation against the background of vaccination carried out over 20 years. Another possible reason for incomplete vaccine-induced protection against HBV resulting in transient subclinical infections and detectable anti-HBc antibodies may be the mismatch between the vaccine HBV genotype/serotype and circulating HBV strains, leading to limited cross-protection [40]. Although this cannot be completely ruled out, the insufficient cross-protection seems not to be the major reason for the presence of anti-HBc in the vaccinated generation observed in our study, as internationally produced HBV-A/ad vaccines and domestically produced HBV-D/ay vaccines are both available and widely used in Russia [41]. Moreover, the prevalence of ad and ay HBV strains has remained stable over the 20 years of mass vaccination, indicating the absence of any serotype-specific selection. The decline in HBV-A genotype prevalence observed in Yakutia was recently confirmed by an analysis of population dynamics [16] but was presumably associated with epidemic patterns in high-risk groups and not with preferential prevention with HBV-A/ad vaccine. As such, the most obvious reason for continued HBV circulation in the vaccinated population is an insufficient level of immunity, probably due to vaccination gaps and insufficient vaccination timeliness. The latter has been reported previously in several regions of Russia [5].

Therefore, the second objective of our study was to evaluate the level of vaccine-induced immunity in the general population and to determine the proportion of subjects who have no routinely measurable vaccine- or infection-induced immunity to HBV. The average anti-HBs detection frequency observed in the generation vaccinated at birth and the age-dependent rate of its decline were comparable to data reported from different parts of the world [42,43]. However, in several study regions (Kaliningrad Region, the Republic of Dagestan, Sverdlovsk Region, and the Tuva Republic), vaccine-induced anti-HBs rates in the vaccinated generation were significantly lower compared to the national average, being the lowest (below 10%) for the 10–19 years age group in the Tuva Republic. Similarly, low anti-HB prevalence rates were reported in the Arctic region of Canada among the 5–9 years age group following primary vaccination [44]. Theoretically, the decreased rates of anti-HBs antibodies in vaccinated generation may be associated with the possible impairment of immune responses by high maternal anti-HBs, as it was demonstrated earlier [45]. However, further studies demonstrated that even high titers of maternal anti-HBs do not inhibit the long-term immunogenicity of Hepatitis B vaccine in infants [46]. Thus, the maternal anti-HBs status may have little impact on the effectiveness of the standard infant vaccination schedule. More likely, the possible reasons for such low levels of vaccine-induced immunity observed in several regions in our study may be low coverage of complete vaccination, or corrupted vaccine immunogenicity due to faults in the vaccine cold storage chain, or some other undetermined factors, including specific human leukocyte antigen (HLA) types possibly associated with antibody response to HBV vaccination [47]. Noteworthy, neonatal HBV vaccination, being the part of national vaccination schedule, is obligatory but voluntary, providing parents the opportunity to refuse to vaccinate their children. Data of the analysis of newborn medical records from two regions of Russia indicated that parental refusal was the most frequent reason of infant HBV non-vaccinated status [5]. However, as no participant medical records were analyzed in current study, and the HBV vaccination status was assumed based on the age of the individual, the true coverage of complete HBV vaccination in study regions and individual participant vaccination status remain unknown. This constitutes a limitation of this study.

Our data on both the age-specific rates of vaccine-induced HBV immunity and the proportion of participants who have no detectable humoral immunity to HBV indicate a paradoxical situation, whereby young adults are better protected from HBV than children despite 20 years of infant immunization. Evidently, the catch-up HBV vaccination in young adults works more efficiently than child vaccination, at least in terms of detectable humoral immunity. Nevertheless, the absence of detectable anti-HBs in those vaccinated at birth does not necessarily indicate a lack of protection, as immunologic memory is believed to provide sufficient protection against HBV infection and associated liver disease [48]. Due to this, the monitoring of anti-HBs levels and booster doses of vaccine is not recommended by WHO for people who have completed the three-dose vaccination course [49]. However, vaccine-induced immune memory may not be enough to protect from subclinical infections in adulthood [50]. Consequently, since 2021, annual anti-HBs screening has been recommended in Russia for healthcare workers, with a booster dose of vaccine administered to those who have an antibody level below 10 mIU/mL [51]. This measure, given that it is aimed at only one group facing occupational risk, may be not sufficient to constrain HBV circulation in populations with a neonatal vaccination program of restricted immunological effectiveness. Further studies are needed to understand the exact reasons for low vaccine-induced HBV immunity in some populations. The low protection rates observed in this study in the generation vaccinated at birth suggest that HBV vaccination quality control should be implemented. This involves determining anti-HBs levels in random samples of vaccinated individuals and analyzing vaccination timeliness and the series of vaccines used for immunization. Moreover, the lower-than-expected rates of vaccine-induced immunity in individuals vaccinated at birth observed in several regions indicates the need for a possible expansion of booster immunization programs in young adults in the near future.

## 5. Conclusions

Twenty years of a neonatal HBV vaccination program, supplemented by catch-up vaccination in adults, has been successful in the vast majority of regions in terms of reducing the frequency of HBV infection to HBsAg detection rates in line with the target prevalence indicator of the WHO elimination plan. However, anti-HBc detection rates observed in the generation vaccinated at birth indicate ongoing HBV circulation, resulting in subclinical transient infections. The absence of vaccine-driven positive selection of immune-escape variants over 20 years of a mass vaccination program, together with low rates of vaccine-induced immunity observed in age groups vaccinated at birth, suggests that gaps in vaccination, and not virus evolution, may be responsible for the continued HBV circulation. To overcome this, a vaccination quality control program, i.e., a “vaccine audit,” needs to be implemented, along with booster immunization for young adults in populations where neonatal HBV vaccination uptake was suboptimal.

## Figures and Tables

**Figure 1 vaccines-11-00430-f001:**
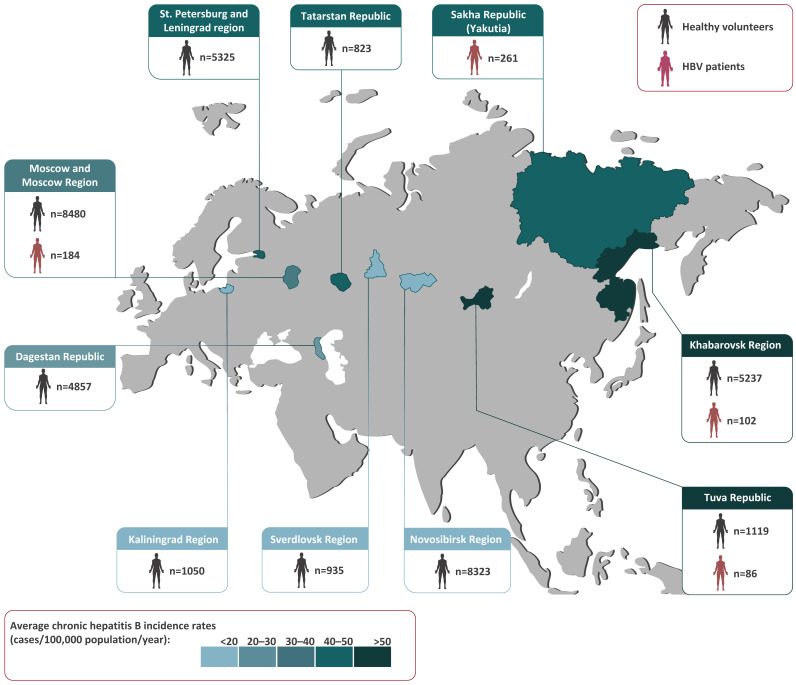
The study regions (in color) shown on a map of Russia alongside the numbers and sources of the samples collected in each region. The colored bar represents the 11-year average chronic Hepatitis B incidence rates (cases per 100,000 population per year) from 2010–2021, based on a federal state surveillance report [4].

**Figure 2 vaccines-11-00430-f002:**
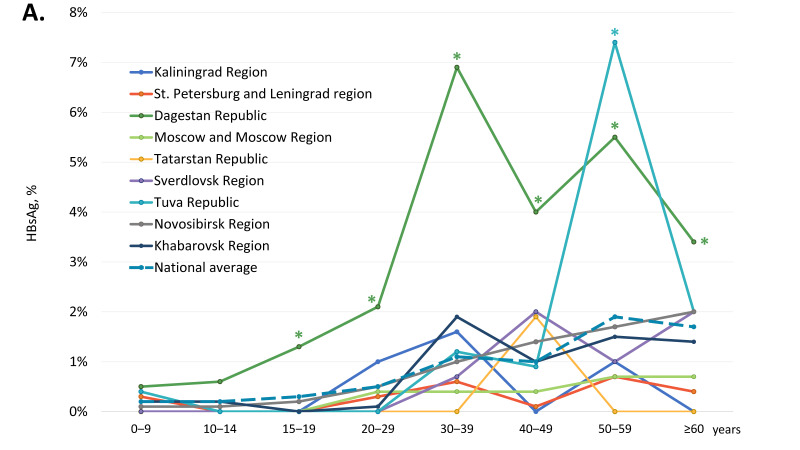
Age-specific HBsAg (**A**) and anti-HBc antibody (**B**) detection rates among healthy volunteers in the study regions. Values that are significantly higher than the national average values (two-tailed *p* < 0.05, Chi-square with Yates’ correction) are marked with an asterisk (*).

**Figure 3 vaccines-11-00430-f003:**
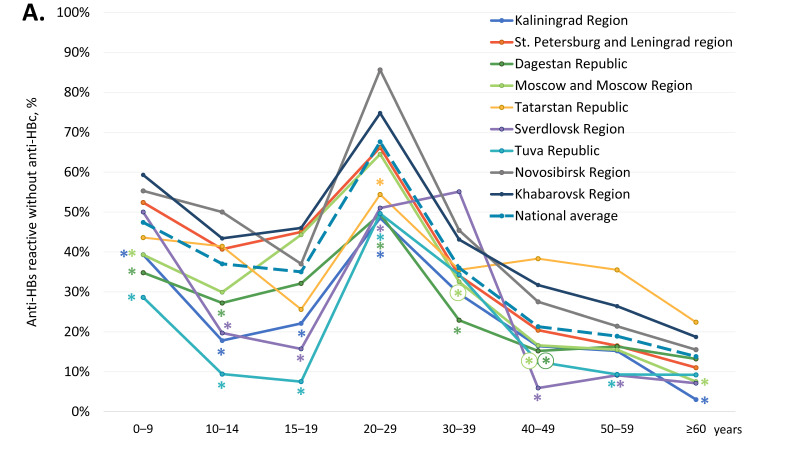
Age-specific proportions of serum samples reactive for anti-HBs but non-reactive for anti-HBc antibodies (**A**), and non-reactive for either anti-HBs or anti-HBc (**B**). Values significantly lower (Panel **A**) or higher (Panel **B**) than the national average values (two-tailed *p* < 0.05, Chi-square with Yates’ correction) are marked with an asterisk (*).

**Figure 4 vaccines-11-00430-f004:**
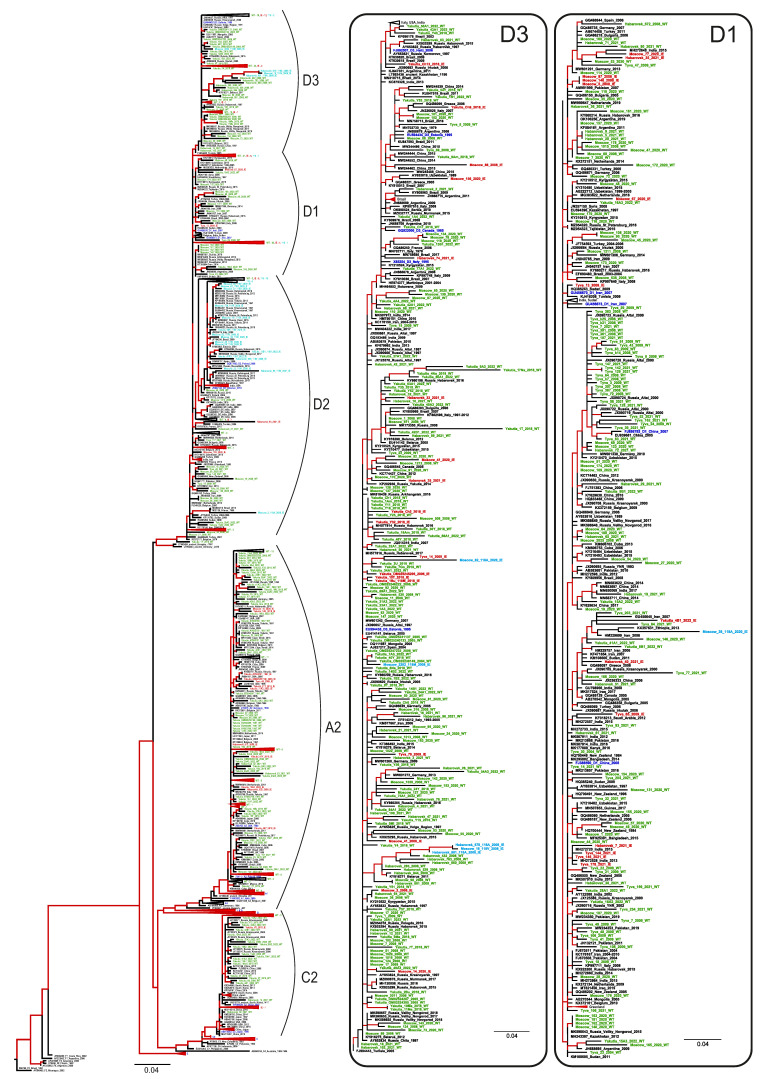
ML phylogenetic tree based on the 640 nt fragment of the HBV S-gene (nucleotide positions 157–796 by reference sequence NC_003977.2). Tree branches with a posterior probability of >90% are shown in red. Clusters containing sequences from this study and representing sub-genotypes D1, D2, D3, A2, and C2 are shown in separate panels. The GenBank database number, the HBV genotype, the country, and the year of isolation are indicated for each sequence. Sequences from this study are shown using the following colors and abbreviations: green (WT)—wild-type, red (IE)—immune-escape HBsAg variants with the exception of T118A/V, light blue (IE)—T118A/V variants, blue—reference sequences for HBV genotypes and sub-genotypes.

**Figure 5 vaccines-11-00430-f005:**
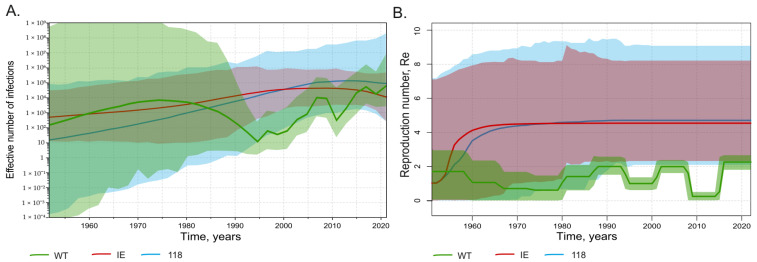
SkyGrid reconstruction (**A**) showing the relationship between the effective number of infections (*y* axis) and chronological time expressed in years (*x* axis), and Birth–Death Skyline reconstruction (**B**) showing the relationship between the reproduction number of infections (*y* axis) and chronological time expressed in years (*x* axis). The curves indicate the mean, and the 95% HPD interval is shown by the shaded area. Graphs for wild-type HBV sequences are shown in green, for immune-escape HBsAg variants except T118A/V in red, and for T118A/V variants in light blue.

**Table 1 vaccines-11-00430-t001:** Average detection rates of HBV markers in healthy volunteers.

Study Region	Reactive for HBsAg, % (95% CI)	Reactive for Anti-HBc, % (95% CI)	Reactive for Anti-HBs, but Non-Reactive for Anti-HBc, % (95% CI)	Non-Reactive for Either Anti-HBs or Anti-HBc, % (95% CI)
Kaliningrad Region	0.4% (0.1–1.0%)	17.1% (15.0–19.5%) *	27.0% (24.4–29.7%) *	57.5% (54.5–60.5%) *
St. Petersburg and Leningrad Region	0.4% (0.2–0.6%)	10.2% (9.4–11.1%)	40.4% (39.1–41.7%)	49.4% (48.1–50.7%) *
Republic of Dagestan	2.4% (2.0–2.9%) *	23.1% (21.9–24.3%) *	29.8% (28.5–31.1%) *	47.1% (45.7–48.5%)
Moscow and Moscow Region	0.4% (0.3–0.5%)	11.3% (10.6–12.0%)	37.5% (36.5–38.5%)	51.1% (50.0–52.2%) *
Republic of Tatarstan	0.2% (0.1–0.9%)	11.5% (9.5–13.9%)	37.1% (33.8–40.4%)	51.0% (47.6–54.4%) *
Sverdlovsk Region	0.6% (0.3–1.4%)	16.5% (14.2–19.0%)	27.7% (24.9–30.7%) *	55.9% (52.7–59.1%) *
Tuva Republic	1.2% (0.7–2.1%)	31.7% (29.1–34.5%) *	22.0% (19.6–24.5%) *	46.0% (43.1–49.0%)
Novosibirsk Region	0.8% (0.6–1.0%)	11.8% (11.1–12.5%)	46.7% (45.6–47.8%)	41.5% (40.4–42.6%)
Khabarovsk Region	0.7% (0.5–0.9%)	14.1% (13.2–15.0%)	47.3% (45.9–48.7%)	38.7% (37.4–40.0%)
National average	0.8% (0.7–0.9%)	14.2% (13.8–14.5%)	39.4% (38.9–39.9%)	46.5% (46.0–47.0%)

* *p* < 0.05 (Chi-square with Yates’ correction) when compared with the national average value.

**Table 2 vaccines-11-00430-t002:** Prevalence of HBV variants carrying amino acid substitutions in HBsAg in positions associated with immune escape.

Study Region	Year of Sample Collection and Number of HBV Sequences, *n*	HBV Genotypes/ Serotypes, *n* (%)	aa Substitutions in HBsAg Associated with Immune Escape					
			aa Substitution *	Number of Sequences Carrying aa Substitution	The Proportion of Sequences with aa Substitution			
					Per HBV Genotype **	Total per Region **	Total per HBV Genotype, Excluding aa Substitutions in Position 118 **	Total per Region, Excluding aa Substitutions in Position 118 **
Moscow Region	2008–2010, *n* = 55	A/*ad*, *n* = 3 (5.5%)	n.d.	0	0/3		0/3	
		D/*ay*, *n* = 52 (94.5%)	T118A/V/M	6/5/1	18/52 (34.6%)	18/55 (32.7%)	6/52 (11.5%)	6/55 (10.1%)
			P120Q	1				
			*T131I/P*	*3/1*				
			*D144E*	*1*				
	2020, *n* = 136	A/*ad*, *n* = 11 (8.1%)	*G130N*	1	1/11		1/11	
		D/*ay*, *n* = 125 (91.9%)	T118A/V	21/11	44/125 (35.2%)	45/136 (33.1%)	12/125 (9.6%)	13/136 (9.6%)
			P120Q	1				
			*Q129H/L/P*	*1/2*				
			*G130R/S*	*1/1*				
			*T131I/N*	*1/1*				
			*M133T/I*	*3*				
			*G145R*	*1*				
Tuva Republic	2008, *n* = 76	A/*ad*, *n* = 2 (2.6%)	*M133T*	*1*	1/2		1/2	
		D/*ay*, *n* = 74 (97.4%)	T118A	13	19/74 (25.7%)	20/76 (26.3%) ▪	6/74 (8.1%)	7/76 (9.2%)
			P120T	1				
			*T131N*	*1*				
			*M133L/K/I*	*1/2*				
			*G145S*	*1*				
	2021, *n* = 49	A/*ad*, *n* = 2 (4.1%)	n.d.	0	0/2		0/2	
		D/*ay*, *n* = 47 (95.9%)	T118A	19	22/47 (46.8%)	22/49 (44.9%)▪	3/47 (6.4%)	3/49 (6.1%)
			P120S	3				
Sakha Republic (Yakutia)	1997–2009, *n* = 58	A/*ad*, *n* = 33 (56.9%)	*T126S*	*2*	4/33 (12.1%)		4/33 (12.1%)	
			*M133T*	*2*				
		C/*ad*, *n* = 2 (3.4%)	n.d.	0	0/2		0/2	
		D/*ay*, *n* = 23 (39.7%)	T118A	2	7/23 (30.4%)	11/58 (19.0%)	5/23 (21.7%)	9/58 (15.5%)
			T118A+P120I	1				
			P120S/T	2				
			*M133T*	*1*				
			*K141Q*	*1*				
	2018–2022, *n* = 231	A/*ad*, *n* = 94 (40.7%)	P120T	1	7/94 (7.4%) ▪▪	36/231 (15.6%)	7/94 (7.4%)	14/231 (6.1%)
			*Q129R*	*1*				
			*Q129P*	*1*				
			*G130N*	*1*				
			*M133T*	*2*				
		C/*ad*, *n* = 21 (9.1%)	*M133I*	*1*	1/21 (4.8%)		1/21 (4.8%)	
		D/*ay*, *n* = 116 (50.2%)	T118A/V	19/4	29/116 (25.0%) ▪▪		6/116 (5.2%)	
			T118A+*Q130R*	1				
			T118M+ P120S	1				
			P120S	1				
			*G130N+T131H*	*1*				
			*T131I/P*	*1/1*				
Khabarovsk Region	2008, *n* = 16	D/*ay*, *n* = 16 (100%)	T118A/V	1/2	4/16 (25%)	4/16 (25%)	0/16 (0%)	0/16 (0%)
	2021, *n* = 65	A/*ad*, *n* = 7 (10.8%)	n.d.	0	0/7		0/7	
		C/*ad*, *n* = 2 (3.1%)	n.d.	0	0/2		0/2	
		D/*ay*, *n* = 56 (86.1%)	T118A/V	7/2	18/56 (32.1%)	18/65 (27.7%)	9/56 (16.1%)	9/65 (13.8%)
			T118A+G130A	1				
			P120S	1				
			T123I	1				
			*T126N+D144E*	*1*				
			*G130S*	*1*				
			*T131N*	*1*				
			*M133S/T*	*3*				

* Amino acid (aa) substitutions with immune-escape phenotype known from literature data, as well as respective numbers are shown in bold; other substitutions in the same aa positions and respective numbers are shown in plain text; aa substitutions within “a” determinant (aa positions 124–147) and respective numbers are shown in italics. ** Combined data are given on aa substitutions both with the known immune-escape phenotype and other substitutions in the same aa positions. n.d. – not detected. ▪ *p* < 0.05 (Fisher’s exact test) in pair-wise comparison between two time points. ▪▪ *p* < 0.05 (Fisher’s exact test) in pair-wise comparison between two HBV genotypes.

## Data Availability

The data presented in this study are available in this article and its Appendix A.

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
