# Peer review of "Post-Vaccination and Post-Infection Immunity to the Hepatitis B Virus and Circulation of Immune-Escape Variants in the Russian Federation 20 Years after the Start of Mass Vaccination"

_vaccines, 2023, doi:10.3390/vaccines11020430_

Round 1

Reviewer 1 Report

The authors have assessed vaccine-and infection-vaccine induced immunity to HBV in a large sample of voluntary recruited 0-95 years old subjects. Prevalence of immune-escape mutants has been also evaluated.

Findings provided lack of novelty, as several worldwide papers have previously explored in more valuable manner this topic. Most important, the study presents pittfalls affecting the validity of the observed findings; lack of some information hampers the assessment of vaccine effectiveness:

-The sample size was not randomly selected, but on voluntary basis; it generates a typical referral bias.

-The neonatal vaccination is just recommended or compulsory by law?

- Which is the neonatal vaccination coverage?

-How vas ascertained a previous HBV vaccination?

Discussion very confusing with several useless paragraphs.

The paper does not stand-up in the present form. I suggest to send the paper to a journal of basic virology, focusing exsclusively on the HBV genome findings.

Finally, it's very hard jusify the presence of the awesome number of cohauthors (27!).

Author Response

We are very grateful to Reviewer for comments and thorough analysis of our paper.

Comment 1: Findings provided lack of novelty, as several worldwide papers have previously explored in more valuable manner this topic. Most important, the study presents pittfalls affecting the validity of the observed findings; lack of some information hampers the assessment of vaccine effectiveness:

Response: Although the numerous studies on changes in HBV prevalence following introduction of universal neonatal vaccination have been published so far, we believe that data from different parts of the world are needed to complete the picture. There are several aspects of novelty in our study. First, data on the transition of HBV epidemic from endemic to non-endemic in Russia following 20 years of vaccination has not been published so far. Second, we applied methodology of seroepidemiology and basic virology to address the issue of ongoing HBV circulation in generation vaccinated at birth. We believe, that our findings represent not the unique situation when HBV prevalence do not fall to zero level despite ongoing vaccination, and uncover the possible reasons for this. Thus, this new data may be useful to understand further efforts needed to combat HBV infection. Below we addressed the information lacked in the original manuscript.

Comment 2: -The sample size was not randomly selected, but on voluntary basis; it generates a typical referral bias.

Response: Indeed, there are two possible approaches for sample collection. First, is the random choice of household/phone numbers with subsequent recruitment of chosen participants. In real world practice this approach is associated with exceptionally high level of refusal to participate the survey, making it often inapplicable for seroepidemiological studies. The second approach that was used in this study, is based on invitation to participate the study the volunteers who is already involved in health care services, such as routine medical checkup, vaccination, etc. Normally, such approach make it possible to recruit the larger number of participants without any significant sample bias. In case of the current study, following exclusion criteria were apply to minimize sample bias regarding risks of HBV infection: liver disease, acute illnesses, a body temperature over 37.1°C, or any surgery, blood transfusion, or treatment with blood products within the three months prior to enrollment in the study. Thus, this sample represent the healthy population, and data obtained based on this sample are not associated with the risk of under- or overestimation of HBV prevalence.

Comment 3: -The neonatal vaccination is just recommended or compulsory by law?

Response: Neonatal HBV vaccination is obligatory, as it is included in national immunization schedule. At the same time, HBV vaccination, as any other vaccination, is voluntary, i.e. parents can refuse to vaccinate their child due to personal reasons. Data of the analysis of newborn medical records from two regions of Russia indicated that parent refusal was the most frequent reason of HBV non-vaccination. We added this point to discussion (lines 555-559 in revised manuscript).

Comment 4: - Which is the neonatal vaccination coverage?

Response: Based on reported data, the reported HBV vaccination coverage was 97% in children under 18, and 72% in adults by 2013. We added this information to Introduction section (lines 74-75 in revised manuscript).

Comment 5: -How vas ascertained a previous HBV vaccination?

Response: No participant medical records were analyzed in our study, and the HBV vaccination status was assumed based on the age of the individual, i.e. those born after 1998 belong to vaccinated generation and should be vaccinated at birth. Thus, the true coverage of complete HBV vaccination in study regions and individual participant vaccination status remain unknown. We indicated this as a limitation of our study.

Comment 6: Discussion very confusing with several useless paragraphs.

Response: We built Discussion section based on several goals of the study. First, we discussed the current HBsAg and anti-HBc antibody prevalence and its changes over time following the vaccination. Next, we discussed possible reasons for observed undoing HBV circulation, such as HBV immune-escape mutants or low vaccine-induced immunity in vaccinated generation. Our data suggest the insignificant role of HBsAg immune-escape variants in maintaining HBV circulation. We concluded that low rates of vaccine-induced immunity observed in age groups vaccinated at birth, suggests that gaps in vaccination, and not virus evolution, may be responsible for the continued HBV circulation. We did our best to avoid the unnecessary general reasoning and to discussed only results obtained. We added to discussion some points about influence of maternal anti-HBs on vaccine immunogenicity. We do hope that revised discussion will be found appropriate.   

Comment 7: The paper does not stand-up in the present form. I suggest to send the paper to a journal of basic virology, focusing exsclusively on the HBV genome findings.

Response: We would like to present our data exactly to Vaccines readers, as we believe that the results of this study are mainly focused on HBV vaccination program outcomes and shortcomings, and may be useful for researchers in different parts of the world who perform research in the field of HBV vaccination program.

Comment 8: Finally, it's very hard jusify the presence of the awesome number of cohauthors (27!).

Response: There are 23 coauthors of this paper. The contribution of every author is indicated in section Author Contributions, and every coauthor fulfills the requirements for the authors of scientific publication. Such a long list of contributors is mainly due to the wide geographic of the study and the large number of samples tested. The researchers who performed the participant recruitment and sample collection are indicated as coauthors with “resources” contribution in this manuscript. Likewise, a large number of samples in this study resulted in a significant proportion of coauthors with “investigation” and “data curation” roles.

Reviewer 2 Report

In this manuscript, the authors conclude that anti-HBc prevalence rates observed in the generation vaccinated at birth indicate ongoing HBV circulation, resulting in subclinical transient infections. The absence of vaccine-driven positive selection of immune-escape variants over 20 years of a mass vaccination program, together with low rates of vaccine-induced immunity observed in age groups vaccinated at birth, suggests that gaps in vaccination, and not virus evolution, may be responsible for the continued HBV circulation. Also, the authors expect that a certain proportion of HBsAg-negative, anti-HBc-positive subjects from this study has acquired occult HBV infection despite having been vaccinated at birth.

Please discuss in more details the influence of maternal antibody against hepatitis B surface antigen on active immune response to hepatitis B vaccine in infants (e.g., PLoS One. 2011; 6(9): e25130. Vaccine. 2008 Nov 11;26(48):6064-7).

Please discuss the lowest rates of vaccine-induced HBV immunity observed in four regions of Russian Federation, Kaliningrad Region, the Republic of Dagestan, Sverdlovsk Region, and the Tuva Republic.

Minor points:

Please indicate the criteria of inclusion into the study.

L. 135. Informed written consent was obtained from all participants “or their parents”. Please specify.

What was the sensitivity of HBV core antigen (anti-HBc), and antibodies to HBsAg (anti-HBs) ELISA kits used in this study? Only the sensitivity of the HBsAg ELISA kit is shown.  

Please indicate whether some of the chronic hepatitis B patients HBV DNA sequences analyzed by the authors prior to 2010 were isolated from the same individuals in the present manuscript.

Author Response

We are very grateful to the Reviewer for the comments and thorough analysis of the manuscript.

Comment 1: Please discuss in more details the influence of maternal antibody against hepatitis B surface antigen on active immune response to hepatitis B vaccine in infants (e.g., PLoS One. 2011; 6(9): e25130. Vaccine. 2008 Nov 11;26(48):6064-7).

Response: We thank the Reviewer for suggested references. We added to discussion the possible influence of maternal anti-HBs antibodies on immunogenicity of hepatitis B vaccine (lines 544-550 in revised manuscript).

Comment 2: Please discuss the lowest rates of vaccine-induced HBV immunity observed in four regions of Russian Federation, Kaliningrad Region, the Republic of Dagestan, Sverdlovsk Region, and the Tuva Republic.

Response: We added to the names of the regions with the lowest rates of vaccine-induced HBV immunity (Kaliningrad Region, the Republic of Dagestan, Sverdlovsk Region, and the Tuva Republic) to discussion, and discussed the possible reasons for such a low vaccine-induced HBV immunity (lines 539-540, 544-550, 555-559 in revised manuscript).

Comment 3: Please indicate the criteria of inclusion into the study.

Response: The inclusion criteria were as follows: Apparently healthy with no symptoms of acute disease at the time of enrollment in the study (either self-reported or parent-reported), and permanently resident in the study regions. We indicated these inclusion criteria in the revised manuscript (lines 119-122).

Comment 4: L. 135. Informed written consent was obtained from all participants “or their parents”. Please specify.

Response:  We added “or their parents” (line 139 in revised manuscript). Indeed, for all participants under 15 years the informed consent was obtained from their parents.   

Comment 5: What was the sensitivity of HBV core antigen (anti-HBc), and antibodies to HBsAg (anti-HBs) ELISA kits used in this study? Only the sensitivity of the HBsAg ELISA kit is shown. 

Response: The diagnostic sensitivity of anti-HBc test was 100% (95% CI: 98.5-100%) according to manufacturer’s data. The limit of detection of anti-HBs test was 2 mIU/mL with linear range 10 to 1000 mIU/mL according to kit specifications. We added these data to revised Methods section (lines 152-154)

Comment 6: Please indicate whether some of the chronic hepatitis B patients HBV DNA sequences analyzed by the authors prior to 2010 were isolated from the same individuals in the present manuscript.

Response: None of the sequences obtained prior to 2010 were isolated from the same patients participated the current study, i.e. data sets “before 2010” and “after 2010” did not contained sequences from the same individuals (lines 176-179 in revised manuscript).

Reviewer 3 Report

This is a well designed and well written manuscript, which faces the relevant issue of the impact of HBV vaccination on viral circulation in Russian Federation. This reviewer has no particular suggestions, excepting minor formal points, which may easily be corrected.

In particular, in the Abstract the meaning of HBsAg, anti-HBc and anti-HBs should be explained, as it is for HBV, and as it is explained in the text. Moreover, in the Abstract and in Materials and Methods (lines 34 and 112, respectively) 36,149 should actually be 36,145, which is the sum of the healthy volunteers participating to the study from the 9 Regions, as reported in Fig. 1. Finally, the lateral legend of Fig. 3 b should not be HBsAg, but anti-HBs and anti-HBc non-reactive, as reported in the Legend under the figure.

Author Response

We are very grateful to Reviewer for comments and thorough analysis of our paper.

Comment 1: In particular, in the Abstract the meaning of HBsAg, anti-HBc and anti-HBs should be explained, as it is for HBV, and as it is explained in the text.

Response: We provided the explanation for HBsAg, anti-HBc and anti-HBs in the Abstract, as well as in the text.

Comment 2: Moreover, in the Abstract and in Materials and Methods (lines 34 and 112, respectively) 36,149 should actually be 36,145, which is the sum of the healthy volunteers participating to the study from the 9 Regions, as reported in Fig. 1.

Response: We double-checked the numbers of samples from each region and found the typo in Fig.1, in the number of samples from Dagestan Republic. The correct number of samples from this region was 4857. This correct number of samples was used in all related calculations. We fixed it in the revised Fig.1. Thus, the correct total number is 36,149, as indicated in Abstract and the main text.

Comment 3: Finally, the lateral legend of Fig. 3 b should not be HBsAg, but anti-HBs and anti-HBc non-reactive, as reported in the Legend under the figure.

Response: Thank you for noticing the mistake. We corrected the name of Y-axis to “Anti-HBs and anti-HBc non-reactive”.

Round 2

Reviewer 1 Report

I confirm my previous negative evaluation even for the revised version. Responses provided by authors are not true and unsatisfactory. In details:

-Reponse 1:Findings on the transition of HBV infection from endemic to non endemic situation in a given area have been largely published (see Alaska, Taiwan, Gambia, Italy, Greece) . The lack of fall to zero of HBV prevalence ( better to say "a remarkable reduction") despite ongoing vaccination has been reported and investigated in several countries (USA , most Western European countries). The authors seem ignore the literature on this topic.

Reponse 2: The presence of a referral bias is confirmed by a voluntary instead of by a random enrollment of the sampled population, as reported by the authors.

Reponse 5: The authors confirm that the previous vaccination status of participants is unknown. It represents a serious flaw rather than a so called limit.

Reponse 6: Discussion continues to be confusing and in several parts useless.

Reponse 7: The author's believe is uncorrect. They have mostly addressed attention to HBV genome, a topic of interest for a journal of basic virology.

Reponse: Even 23 cohauthors is an unjustifiable number.

Author Response

We are very grateful to Reviewer for the thorough analysis of our paper, although we cannot agree with all comments made by Reviewer.  

Comment to Response 1: Findings on the transition of HBV infection from endemic to non endemic situation in a given area have been largely published (see Alaska, Taiwan, Gambia, Italy, Greece) . The lack of fall to zero of HBV prevalence ( better to say "a remarkable reduction") despite ongoing vaccination has been reported and investigated in several countries (USA , most Western European countries). The authors seem ignore the literature on this topic.

Response: In no case we intended to ignore literature on changes in HBV prevalence in different counties. We added the respective part to Discussion (lines 451-454 and 463-468). In our previous response we just stressed that data on this topic are necessary from different territories, and our study combine data on changes in HBV prevalence against the background of vaccination from regions of different baseline endemicity level.

Comment to Response 2: The presence of a referral bias is confirmed by a voluntary instead of by a random enrollment of the sampled population, as reported by the authors.

Response: We cannot agree with Reviewer that enrollment of voluntary participants in our study results in biased data on HBV prevalence. In our response we did not confirmed the referral bias, but stated that approach used in our study was based on invitation to participate the study the volunteers who is already involved in health care services, such as routine medical checkup, vaccination, etc. Normally, such approach make it possible to recruit the larger number of participants without any significant sample bias. The exclusion criteria applied at enrollment ensure the conduction of the study among conditionally healthy population. Thus, data obtained based on this sample are not associated with the risk of under- or overestimation of HBV prevalence.

Comment to Response 5: The authors confirm that the previous vaccination status of participants is unknown. It represents a serious flaw rather than a so called limit.

Response: We have not analyzed the effectiveness of HBV vaccination in our study. In our study, we assessed the HBV circulation in different age groups of general population against the background of current vaccination policy and the impact of HBV variants on it. In this regard, the vaccination status of participants is not the information crucial for conclusions of the study. Therefore, we do not consider the lack of data on previous vaccination status of participants as a serious flaw.

Comment to Response 6: Discussion continues to be confusing and in several parts useless.

Response: We could not identify the confusing or useless parts in Discussion section. Each paragraph in Discussion is devoted to particular result and its interpretation. First, the current HBsAg and anti-HBc antibody prevalence and its changes over time following the vaccination. Second, the possible reasons for observed undoing HBV circulation are discussed. Third, HBV immune-escape mutants and one particular, highly prevalent variant are discussed. Fourth, the rates of vaccine-induced immunity in vaccinated generation are discussed. Finally, we concluded that low rates of vaccine-induced immunity observed in age groups vaccinated at birth, suggests that gaps in vaccination, and not virus evolution, may be responsible for the continued HBV circulation. We would be grateful to Reviewer for specifying the useless and confusing parts in Discussion.  

Comment to Response 7: The author's believe is uncorrect. They have mostly addressed attention to HBV genome, a topic of interest for a journal of basic virology.

Response: We cannot agree with reviewer on this matter. Our study is focused on relationship between HBV circulation and herd immunity to HBV, both vaccine- and infection-induced. This topic falls into the scope of Vaccines and its Special Issue "Dynamic Models in Viral Immunology" in particular.

Comment to Response: Even 23 cohauthors is an unjustifiable number.

Response: In this matter, we only can repeat the previous response: Every coauthor fulfills the requirements for the authorship of the scientific publication. Such a long list of contributors is mainly due to the wide geographic of the study and the large number of samples tested. The researchers who performed the participant recruitment and sample collection are indicated as coauthors with “resources” contribution in section Author Contributions at the end of the manuscript. Likewise, a large number of samples in this study resulted in a significant proportion of coauthors with “investigation” and “data curation” roles.

Reviewer 2 Report

My questions and comments were satisfactorily answered. I do not have further questions

Author Response

Comment: My questions and comments were satisfactorily answered. I do not have further questions

Response: We are very grateful to Reviewer for the thorough analysis of our paper and the positive comment.

Round 3

Reviewer 1 Report

Reponse 1. I repeat that the study lack of novelty, as this topic has been largely explored in the literature.

Reponse 2. Criteria for selecting a target population in a cross-sectional survey are likely unclear to the authors. It is well known that in a cross-sectional survey representativeness is the most important point to achieve, otherwise findings are likely affected by selection and referral biases. Recruitment of the target population by a random sampling procedure represents the only manner to obtain representativeness. Recruitment by invitation lack of representativeness. This is an important limit in the study design, causing a lack of validity in the obseved findings.

Response 5. Ascertainment of the vaccination status of subjects is a crucial point to define the true HBV circulation, as vaccinated people are no longer at risk of infection. Indeed, lack of information regarding vaccination status is a serious flaw.

Response 6. I confirm my previos negative evaluation of the Discussion.

Reponse 8, Once again, the authors try to justify the excessive number of coauthors, despite it is really awesome, In these cases a given number of coauthors are enclosed in the "collaborating group"

Author Response

Comment 1. I repeat that the study lack of novelty, as this topic has been largely explored in the literature.

Response: In our manuscript, we present the data on this topic from Russia that were absent so far in literature. These data include both data on HBV seroepidemilology and HBV molecular evolution under the pressure of mass vaccination from different geographically remote parts of the country that are rare in published literature.

Comment 2. Criteria for selecting a target population in a cross-sectional survey are likely unclear to the authors. It is well known that in a cross-sectional survey representativeness is the most important point to achieve, otherwise findings are likely affected by selection and referral biases. Recruitment of the target population by a random sampling procedure represents the only manner to obtain representativeness. Recruitment by invitation lack of representativeness. This is an important limit in the study design, causing a lack of validity in the observed findings.

Response:  We have not referred our study as a cross-sectional survey. We agree that although sample collection is "surrogate" (not in a bad sense) to a cross-sectional survey, they are useful; especially, if no other data are available from the region. Using voluntary participants as a surrogate to a randomized cross-sectional study, which is not affordable due to often-poor quality of medical records and low level of uptake from participants, is a surrogate survey. Will it give you an idea about the frequency of disease there? Yes. Would you be able to say that is "prevalence"? No. Thus, we modified the manuscript to avid usage of “prevalence” term that may be confusing considering that our study was not a cross-sectional survey. We used “detection rate” instead throughout the revised manuscript.

Comment 3. Ascertainment of the vaccination status of subjects is a crucial point to define the true HBV circulation, as vaccinated people are no longer at risk of infection. Indeed, lack of information regarding vaccination status is a serious flaw.

Response: We agree that assessment of the vaccination status of subjects can provide a useful information on a proportion of people who are no longer at risk of infection. But, it works well only when the quality of medical records that are the source of information, are of significant quality. Unfortunately, in general the medical records in Russia contain no or little information on history of vaccinations. In this case, the assessment of vaccination status of study participant is largely depend on participant’s self-reported status, which is often not reliable. Due to this limitation, we relied on results of testing for serum anti-HBs as the only reliable indicator of immunity to HBV. Moreover, the official statistics on HBV vaccination coverage (up to 97% in children under 18) that is based on the data on the numbers of vaccination from each region, has a little correlation with actual rates of anti-HBs positivity in vaccinated generation in several studied regions. That is why we stressed in Conclusion the need for vaccination quality control program implementation.

Comment 4. I confirm my previous negative evaluation of the Discussion.

Response: We changed the “prevalence rates” to “detection rates” to avoid confusion related to study design. We kindly ask Reviewer to be specific in evaluation of Discussion and indicate other confusing or useless sentences, if any.

Comment 5. Once again, the authors try to justify the excessive number of coauthors, despite it is really awesome, In these cases a given number of coauthors are enclosed in the "collaborating group"

Response: We moved seven coauthors from coauthor list to a collaborating group, as suggested.